# Metabolomic Signature Discriminates Normal Human Cornea from Keratoconus—A Pilot GC/MS Study

**DOI:** 10.3390/molecules25122933

**Published:** 2020-06-25

**Authors:** Anna Wojakowska, Monika Pietrowska, Piotr Widlak, Dariusz Dobrowolski, Edward Wylęgała, Dorota Tarnawska

**Affiliations:** 1European Centre for Bioinformatics and Genomics, Institute of Bioorganic Chemistry Polish Academy of Sciences, Noskowskiego12/14, 61-704 Poznan, Poland; astasz@ibch.poznan.pl; 2Maria Sklodowska-Curie National Research Institute of Oncology, Gliwice Branch, Wybrzeze Armii Krajowej 15, 44-102 Gliwice, Poland; monika.pietrowska@io.gliwice.pl (M.P.); piotr.widlak@io.gliwice.pl (P.W.); 3Department of Ophthalmology & Tissue and Cells Bank, St. Barbara Hospital, Trauma Center, Plac Medyków 1, 41-200 Sosnowiec, Poland; dardobmd@wp.pl; 4Chair and Clinical Department of Ophthalmology, Division of Medical Science in Zabrze, Medical University of Silesia, Panewnicka 65, 40-760 Katowice, Poland; wylegala@gmail.com; 5Department of Ophthalmology, District Railway Hospital, Panewnicka 65, 40-760 Katowice, Poland; 6Faculty of Science and Technology, Silesian Center for Education and Interdisciplinary Research, University of Silesia, 75 Pułku Piechoty 1A, 41-500 Chorzów, Poland

**Keywords:** keratoconus, cornea, ocular metabolomics, GC/MS profiling

## Abstract

The molecular etiology of keratoconus (KC), a pathological condition of the human cornea, remains unclear. The aim of this work was to perform profiling of metabolites and identification of features discriminating this pathology from the normal cornea. The combination of gas chromatography and mass spectrometry (GC/MS) techniques has been applied for profiling and identification of metabolites in corneal buttons from 6 healthy controls and 7 KC patients. An untargeted GC/MS-based approach allowed the detection of 377 compounds, including 46 identified unique metabolites, whose levels enabled the separation of compared groups of samples in unsupervised hierarchical cluster analysis. There were 13 identified metabolites whose levels differentiated between groups of samples. Downregulation of several carboxylic acids, fatty acids, and steroids was observed in KC when compared to the normal cornea. Metabolic pathways associated with compounds that discriminated both groups were involved in energy production, lipid metabolism, and amino acid metabolism. An observed signature may reflect cellular processes involved in the development of KC pathology, including oxidative stress and inflammation.

## 1. Introduction

Keratoconus (KC) is among the most common indications for corneal transplantation [1,2,3]. This is an eye ectatic disorder characterized by the progressive corneal thinning, protrusion, and irregular myopic astigmatism, which in advanced cases can lead to severe visual distortions. KC is a serious clinical problem worldwide. The prevalence of the disease in the general population is estimated at 50 to 250 cases per 100,000 individuals [4,5], and even higher in some regions of Asia [6,7]. KC is primarily diagnosed in youths and young adults, hence it is the most common indication for corneal transplantation in people under 40 years of age [2,3,8]. Despite a great deal of research, the pathogenesis of KC is still poorly understood. Most likely, the condition results from the interaction of multiple factors including genetic and environmental ones. Among proven intrinsic mechanisms involved in etiology of KC, there are alterations in enzymes activity (e.g., lysyl oxidase, matrix metalloproteinases) that, by attenuation of covalent bonds between collagen fibrils and disorganization of the extracellular matrix, can lead to deterioration of biomechanical and optical properties of the cornea [9,10,11]. What is more, several reports are showing that corneal components involved in protection against oxidative stress damage are changed in this disorder. The increased levels of oxidative stress markers and the decreased antioxidant defenses in KC corneas can indicate that oxidative stress might be involved in the development of this pathology [9,12,13,14,15,16].

Recent studies point to the metabolic changes as a key to understand the pathogenesis of eye diseases [17,18,19]. The pathologic processes occurring in corneal tissue can be reflected in the metabolome changes, causing the increase or decrease of particular metabolites [20]. Metabolomic indicators of oxidative stress, such as glutathione, are considered as one of the major factors involved in the development of corneal diseases [21]. Furthermore, changes in the lipid composition of eye tissue can be responsible for the modulation of responses to inflammation and oxidative stress occurring in ophthalmic pathologies [22].

Despite the growing number of translational and laboratory studies proving the role of oxidative stress and inflammation in KC, it should be emphasized that probably not all forms of KC can be definitively linked to inflammatory metabolites [23]. Inflammation alone is clearly not sufficient to induce KC, because many patients with severe keratitis do not develop KC. Another inconsistency that is pointed out is the lack of histological and clinical features of inflammation, such as cell infiltration and neovascularization in keratoconic corneas [24]. The complex interactions between genetic predisposition to abnormalities of any of inflammatory components and environmental triggers (eye rubbing, wearing contact lenses, or exposure to ultraviolet light) are likely to be the key to understanding the role of inflammatory mediators and its diverse contribution in the development of various subtly different phenotypes of KC [25,26].

In recent years several groups have investigated the metabolome composition of keratoconic cornea tissue [27] as well as the aqueous humor [28] and tear film of KC patients [20]. The metabolome composition of the human cornea has been studied with the use of different NMR spectroscopy techniques, including ^31^P NMR [29] and HR-MAS ^1^H NMR [27]. Mass spectrometry methods, mainly coupled to liquid chromatography (LC/MS), were applied for metabolomics studies of the human lens [30], tear fluid [21], or aqueous humor [20]. However, the knowledge of the metabolome composition of the human cornea in specific pathogenic conditions remains very limited. Moreover, there are some limitations of above mentioned analytical technics, including low sensitivity of NMR and narrowing of LC/MS analytical capabilities only to polar compounds. In the present study, the combination of gas chromatography and mass spectrometry (the GC/MS approach) has been applied for the first time to the profiling of the metabolome of keratoconus, which reveals metabolomics signature discriminating this pathology and the healthy human cornea.

## 2. Methods

### 2.1. Ethics Statement

The study protocol adhered to the tenets of the Declaration of Helsinki and was approved by the Bioethical Committee of the Maria Skłodowska-Curie Institute—Oncology Center, Branch Gliwice, Poland (permit number KB/430-01/14). Written informed consent was obtained from all patients.

### 2.2. Tissue Collection

Seven corneal buttons 7.5–8.25 mm in diameter derived from patients who underwent penetrating corneal transplantation due to keratoconus (KC) were used for the analysis. Six normal cornea samples from the donor material not suitable for transplantation were obtained from the Eye Tissue and Cells Bank, St. Barbara Hospital, Sosnowiec, Poland, and used as healthy controls (HC). Corneal buttons trephined from normal corneas were fully transparent and 8.0 mm in diameter. Because altered lipid retention was reported in diabetic corneas [31], patients with an abnormal lipogram and patients with diabetes were excluded from the study. Moreover, to eliminate possible age-related differences we included samples derived from individuals with a similar age in both groups: KC patients aged 42 to 59 years and HC donors aged 40 to 69 years. All surgeries were performed in the Clinical Department of Ophthalmology, Railway Hospital in Katowice, Poland. All KC eyes included in the study were in an advanced stage of the disease, with high refractive errors and severe loss of visual acuity, but no scarring or a history of acute hydrops. Healthy corneas were stored in a synthetic corneal storage medium (Eusol-C, Alchimia, Padova, Italy) for 5 to 8 days before they were qualified as not suitable for transplantation and used for analyses. Therefore, immediately after excision KC buttons were placed in Eusol-C for seven days before further preparation.

### 2.3. Sample Processing and Metabolite Extraction

The buttons were rinsed twice in PBS and fixed in buffered formalin (3.7% formaldehyde in 10 mM phosphate buffer, pH 7.4) for 24 h at room temperature, then frozen and stored at −80 °C until analysis. The frozen tissue samples were ground in liquid nitrogen in precooled adaptors for 45 s at 30 Hz frequency using a ball mill MM400 (Retsch, Germany). Pulverized tissue was homogenized in 250 μL mixture of 80% MeOH using a vortex mixer for 5 min and placed in an ultrasonic bath for 10 min. The resulting mixture was centrifuged for 10 min at 23,000× *g* in 4 °C. The supernatant was transferred to a new tube (polar fraction) and the pellet was suspended in 250 μL of a mixture of CH_2_Cl_2_:MeOH (3:1 *v*/*v*) and processed as described above to generate the non-polar fraction. Both polar and non-polar fractions were combined, transferred to a new tube and evaporated in a SpeedVac concentrator. The dried extract was then derivatized with 25 μL of methoxyamine hydrochloride in pyridine (20 mg/mL) at 37 °C for 90 min with agitation. The second step of derivatization was performed by adding 40 μL of N-Methyl-N-(trimethylsilyl)trifluoroacetamide (MSTFA) and incubation at 37 °C for 30 min with agitation. Samples were subjected to GC/MS analysis directly after derivatization.

### 2.4. GC/MS Analysis

The GS/MS analysis was performed using an Agilent 7890A gas chromatograph (Agilent Technologies) connected to a Pegasus 4D GCxGC-TOFMS mass spectrometer (Leco). A DB-5 bonded-phase fused-silica capillary column (30 m length, 0.25 mm inner diameter, 0.25 μm film thickness) (J&W Scientific Co., Folsom, CA, USA) was used for separation. The GC oven temperature program was as follows: 2 min at 70 °C, raised by 8 °C/min to 300 °C and held for 16 min at 300 °C. The total time of the GC analysis was 46.75 min. Helium was used as the carrier gas at a flow rate of 1 mL/min. One microliter of each sample was injected in the splitless mode. The initial injector temperature was 20 °C for 0.1 min and after that time raised by 600 °C/min to 350 °C. The septum purge flow rate was 3 mL/min and the purge was turned on after 60 s. The transfer line and ion source temperatures were set to 250 °C. In-source fragmentation was performed with 70 eV energy. Mass spectra were recorded in the mass range 35–650 m/z.

### 2.5. Analysis of Spectral Data

Data acquisition, automatic peak detection, mass spectrum deconvolution, retention index calculation, and library search were done by the Leco ChromaTOF-GC software (v4.51.6.0). To eliminate retention time (Rt) shift and to determine the retention indexes (RI) for each compound, the alkane series mixture (C-10 to C-36) was injected into the GC/MS system. The metabolites were automatically identified by library search (Replib, Mainlib, Fiehn library); the analyte was considered as identified when it passed the quality threshold: Similarity index (SI) above 700 and matching retention index ±10. Identified artifacts (alkanes, column bleed, plasticizers, MSTFA, and other used reagents) were excluded from further analyses. To obtain accurate peak areas for the deconvoluted components, unique quantification masses for each component were specified and the samples were reprocessed. The obtained profiles were normalized against the sum of the chromatographic peak area (using the total ion chromatograph (TIC) approach—relative quantification).

### 2.6. Statistical and Chemometric Analyses

Quantitative analysis was performed for metabolites identified in at least 60% of samples in each group; the nearest neighbor method was used for missing data imputation. The data were log-transformed. Differences in the abundances of metabolites among two groups were tested using the t-test, Welch test, or Mann–Whitney U test, depending on the normality distribution and homoscedasticity of the analyzed data. The results were presented in arbitrary units as mean abundance after normalization and transformation. Principal Component Analysis (PCA) and hierarchical cluster analysis (HCA) based on the Euclidean distance method were performed to illustrate similarities between samples. All statistical calculations and multivariate data analyses were performed using the Perseus software (http://www.coxdocs.org/doku.php?id=perseus:start). Metabolomic pathways were identified using the Metabolite Set Enrichment Analysis (MSEA) (http://www.msea.ca/MSEA/faces/Home.jsp), which is the metabolomic version of the Gene Set Enrichment Analysis (GSEA) approach. A list of selected metabolites was used as an input for the Over-Representation Analysis (ORA) algorithm, which was implemented using the hypergeometric test to evaluate the over-representation of a particular metabolite set; fold-enrichment values and one-tailed *p*-values corrected for multiple testing were provided.

## 3. Results

The metabolome profiling was performed for 7 samples of keratoconus (KC) and 6 samples of the healthy cornea (healthy controls—HC) using the GC/MS technique. To the best of our knowledge, this is the first application of GC/MS-based protocol (adapted from Wojakowska et al. [32]) in the profiling of metabolites from human corneal buttons. In general, there were 46 unique metabolites identified and quantitated in analyzed samples, which included amino acids, carboxylic acids, fatty acids and their esters, sugars and sugar alcohols, sterols and others (the complete list of identified compounds is presented in Appendix A). Moreover, there were 331 compounds (analytes) detected that could not be identified with the implemented approach. All detected metabolites (both identified compounds and unidentified analytes) were used to perform an unsupervised comparison of samples from both groups. The principal component analysis showed some clustering of samples from both groups, which resulted in the separation of KC and HC samples (Figure 1A). Similarly, the hierarchical cluster analysis also showed the clustering of samples and separation of both groups (Figure 1B). This result indicated the presence of metabolites with abundances significantly different between HC and KC samples, which was further analyzed using supervised tools.

There were 13 identified metabolites whose levels differed significantly (*p* < 0.05) between both types of the cornea (Table 1). In general, 3 carboxylic acids (namely benzoic, glycolic, and succinic) and 6 fatty acids (namely linoleic, myristic, palmitic, pentadecanoic, stearic, and *trans*-13-octadecenoic) were downregulated in keratoconus. Similarly, cholesterol, cholesta-3,5-diene (2 isomers) and hexadecanol were downregulated in the pathological cornea. On the other hand, phosphoric acid was upregulated in KC. Moreover, there were 3 metabolites detected only in one group of samples: gluconic acid and cholesterol propionate were absent in HC samples while petroselenic acid was absent in KC samples (Table 1).

In the next step, 13 metabolites indexed in the Human Metabolome Database (HMDB) that showed significantly different abundances between keratoconus and healthy cornea were subjected to the metabolite set enrichment analysis, which allowed to identify metabolic pathways associated with these discriminatory compounds (Figure 2 shows the top 20 of such pathways). We found that metabolic pathways associated with compounds showing different abundances between HC and KC samples included processes connected with energy production (Warburg effect, tricarboxylic acid cycle, ketone body metabolism, and mitochondrial electron transport chain) and lipid metabolism (fatty acid, glycerolipids and steroid biosynthesis, carnitine synthesis). Moreover, pathways involved in the metabolism of amino acids (Arg, Pro, Val, Leu, Ile, and Glu metabolism) were over-represented among processes associated with compounds that discriminated against different states of the cornea.

## 4. Discussion

Until now, only a few studies concerning metabolome profiling of human cornea have been reported. The molecular composition of this tissue was analyzed using NMR spectroscopy by groups of Greiner [29,33] and Kryczka [16,34,35], yet the number of identified and quantified compounds was rather low due to the limitations of this approach, which is less sensitive than MS and needs more amount of sample for analysis. Moreover, due to the limited availability of corneal tissue for research, many recent LC/MS-based studies on the cornea metabolism and the differences between healthy and keratoconus cornea have been necessarily carried out based on the analysis of tears [21,36], aqueous humor [20], saliva [37] and cell lines [22,38]. In this work, the GC/MS approach was introduced for the first time to the metabolomics profiling of human cornea using corneal buttons. GC/MS-based approach including derivatization of the sample allows analysis of a wide range of compounds (polar and nonpolar). However, the strength of our study is directly related to its limitations, which is the small number of samples tested.

Major classes of detected metabolites were fatty acids, sterols (cholesterol derivatives) and their esters, carboxylic acids (e.g., citric, succinic, and oxalic acids) as well as sugars and their derivatives (glucose, gluconic acid, sorbitol, and myo-inositol). Moreover, because of the limitations of untargeted profiling implemented in the current study, a large number of detected compounds remained unidentified, a problem frequently reported in this type of study [36]. It is worth noting that several unidentified compounds also showed significantly different abundances between samples of the healthy and pathological cornea. Therefore, differences in the molecular composition of these two cornea conditions might be further studied using alternative metabolomics approaches.

The crucial problem in analogous studies is to distinguish the differences in metabolite composition between the normal and diseased tissue from those resulting from post-mortem chemical processes. All normal cornea samples used in our study derived from corneal buttons were harvested up to 3 h after death. This is due to the Eye Bank procedure, according to which the corneas are obtained either at the time of multi organ donation, or soon after two hours have passed since death, in hospital setup. Such an early harvesting time allowed to exclude post-mortem metabolic changes, especially since they proceed very slowly compared to those occurring in the blood or even in the aqueous humor [20].

The molecular pathogenesis of keratoconus is poorly understood. However, it has been suggested that oxidative stress and lipid peroxidation may be involved in this pathological state of the cornea [16]. Oxidative stress in the cornea stimulates the generation of pro-inflammatory cytokines, proteolytic enzymes, and enzymes responsible for the generation of nitric oxide. Accumulation of reactive oxygen species (ROS) and nitric oxide can lead to cell membrane damage by lipid peroxidation and release of cytotoxic aldehydes [10]. Immunohistochemical studies by Buddi et al. revealed the presence of markers of oxidative stress (nitrotyrosine) and lipid peroxidation (malondialdehyde) as well as the expression of endothelial nitric oxide synthase in KC cornea [13]. Moreover, studies revealed changes in the antioxidant status of KC and reported metabolites that are indicators of response to oxidative stress (e.g., glutathione) [10,28,39]. Furthermore, alteration of phosphate metabolism [29] and metabolites associated with the tricarboxylic acid cycle or the Krebs cycle (TCA) and urea cycle [21] were also reported in KC patients.

Here we revealed a few metabolites whose patterns discriminated normal healthy cornea from keratoconus, which included certain carboxylic acids, fatty acids, and sterols markedly downregulated in the pathological state of the tissue. Fatty acids downregulated in KC included saturated (stearic, palmitic, myristic, and pentadecanoic acid) and unsaturated (linoleic and *trans*-13-octadecenoic acid) compounds. The fatty acids predominantly present within the human cornea are oleic, palmitic, and stearic acids [40]. Lipids (along with fatty acids) constitute one of the main components of human cornea and play important roles in complex processes associated with pro- and anti-inflammatory reactions as well as tissue damage and repair. They can act as second messengers for an epidermal growth factor (e.g., lipoxygenase (LOX) derivatives from arachidonic acid) to promote the proliferation and repair of corneal tissue or as mediators of platelet-activating factor (PAF) and cyclooxygenase-2 (COX2) that contribute to tissue damage and neovascularization [41]. Linoleic acid is involved in the biosynthesis of arachidonic acid and prostaglandins (PGs), as well as thromboxane (TXAs) and lipoxygenase (LOX) derivatives, whose levels increase after corneal injury [42]. Moreover, recent studies revealed that polyunsaturated fatty acids (PUFAs), including linoleic acid, play an important role in the reparative processes of the cornea due to their anti-inflammatory effects [43]. Carboxylic acids are another group of compounds discriminating KC and normal cornea. Gluconic acid, here detected exclusively in KC samples, is an intermediate in the pentose phosphate pathway (PPP), essential for the production of NADPH required for the synthesis of fatty acids and lipids [44]. Recent studies showed that dysregulation of the pentose phosphate pathway metabolites, including gluconic acid, in diabetic retinopathy (DR) provided further links to oxidative stress. Therefore, gluconic acid, as a product of glucose oxidation, could be an indicator of oxidative stress in KC. On the other hand, downregulation of succinic acid was observed in KC. Succinic acid plays an important role in TCA (tricarboxylic acid) cycle, the dysregulation of which is associated with changes in energy requirements. In the cornea, the TCA cycle is the key source of energy required to maintain corneal transparency and cellular activity [45]. Under the healthy state, the TCA cycle is relatively quiet; however, it becomes activated upon injury. Recent studies revealed that succinic acid alterations are associated with various retinal diseases [18] and TCA metabolites are highly altered in KC patients [21].

In general, metabolites whose levels differentiated normal and pathological cornea were associated with the production of energy and lipid metabolism. Moreover, identified metabolites were associated with response to oxidative stress, inflammation, and tissue damage. Hence, our observation confirmed previous reports that suggested the role of oxidative stress and pro-inflammatory processes in the development of keratoconus [21,37,38,46].

## 5. Conclusions

The presented work is the first application of the GC/MS-based metabolomic profiling that demonstrated differences in metabolite composition between the keratoconus and normal human cornea. The obtained metabolomic signatures indicate that oxidative stress and inflammatory reactions are involved in the development of this corneal pathology. However, further in-depth metabolomics study that will address a complete ocular metabolome is required to decipher the key metabolic changes crucial for the etiopathology of the keratoconus.

## Figures and Tables

**Figure 1 molecules-25-02933-f001:**
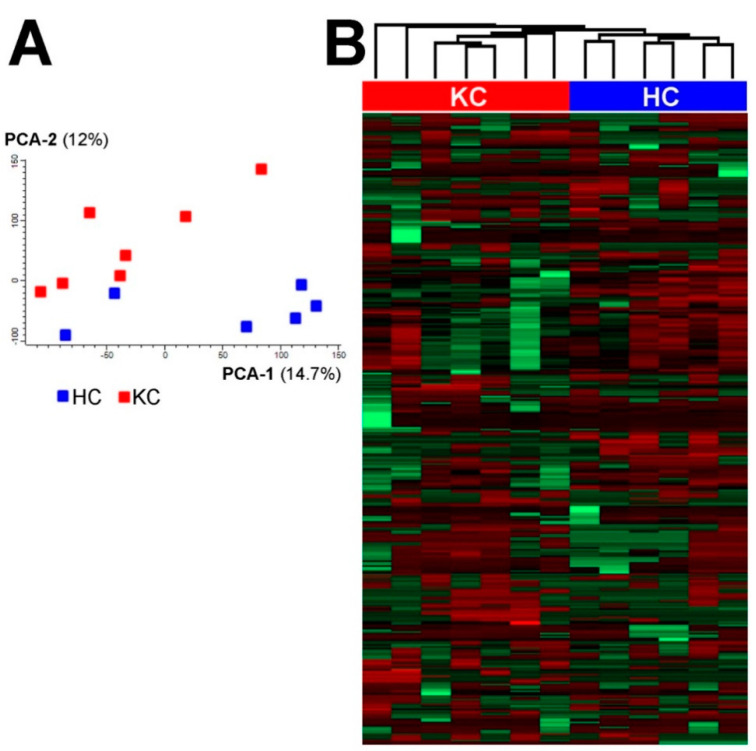
Separation of normal (HC) and keratoconus (KC) samples of human cornea samples by the principal component analysis (**A**) and hierarchical cluster analysis (**B**).

**Figure 2 molecules-25-02933-f002:**
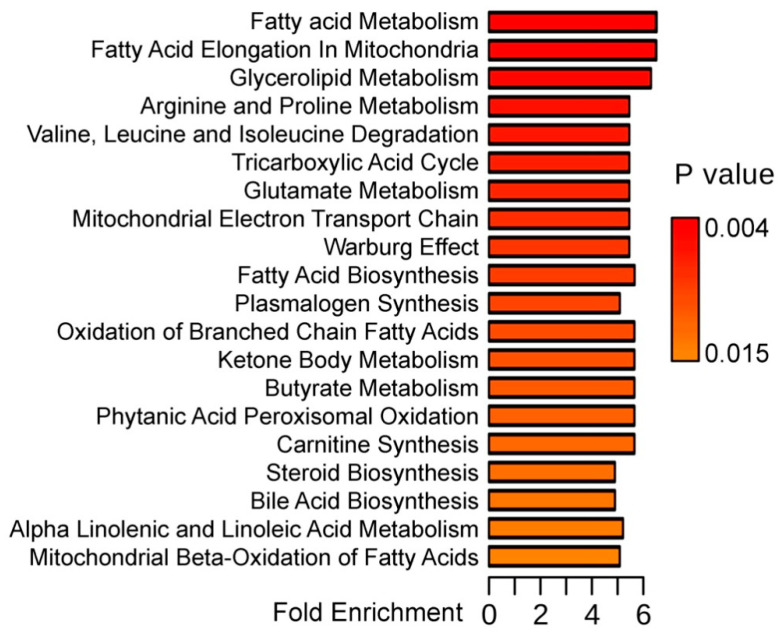
Metabolic pathways associated with 13 metabolites showing significantly different abundances between keratoconus and healthy cornea. Presented are over-representation (fold enrichment) and its statistical significance (uncorrected *p*-value) of metabolic pathway estimated by the metabolite set enrichment analysis (MSEA); the top 20 pathways are represented.

**Table 1 molecules-25-02933-t001:** Metabolites that differentiate the healthy cornea (HC) from keratoconus (KC).

Metabolite Name	Class	HC Mean Abundance	KC Mean Abundance	HC/KC	
				FC	*p*-Value
Benzoic acid	Carboxylic acids	180	119	1.51	0.0338
Glycolic acid	Carboxylic acids	6.18	2.65	2.33	0.0258
Succinic acid	Carboxylic acids	771	536	1.44	0.0096
Gluconic acid	Carboxylic acids	N.D.	0.56	-	-
Linoleic acid	Fatty acids	103	38.8	2.65	0.0144
Myristic acid	Fatty acids	175	112	1.56	0.0197
Palmitic acid	Fatty acids	7240	4250	1.70	0.0042
Pentadecanoic acid	Fatty acids	122	74.3	1.64	0.0163
Stearic acid	Fatty acids	7440	4810	1.55	0.0159
*trans*-13-Octadecenoic acid	Fatty acids	149	83.3	1.79	<0.0001
Petroselinic acid	Fatty acids	256	N.D.	-	-
Cholesta-3,5-diene—isomer 1	Sterols	83.6	17.6	4.75	0.0032
Cholesta-3,5-diene—isomer 2	Sterols	421	85.5	4.92	0.0016
Cholesterol	Sterols	5680	1960	2.90	0.0218
Cholesterol propionate	Sterol esters	N.D.	0.12	-	-
Hexadecanol	Alcohols	109	76.3	1.43	0.0147
Phosphoric acid	Others	56,900	67,700	0.84	0.0228

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
