# Peer review of "Metabolomic Signature Discriminates Normal Human Cornea from Keratoconus—A Pilot GC/MS Study"

_molecules, 2020, doi:10.3390/molecules25122933_

Round 1

Author Response

Reviewer 1

Q1) In the introduction the authors need to explain what new does the GC/MS approach offer over NMR and LC/MS?  They have attempted to elaborate this on the discussion however it needs more explanation.

A1) The corrections were made in the text - lines 79-81 (Introduction) and 211-212; 217-218 (Discussion).

Q2) Why did the authors decide to combine the polar and non-polar fractions prior to evaporation in the speed vac?  How many phases formed after combing the two fractions?  I assume two.  Did this have an effect on efficiently evaporating the sample?  Why not keeping the two fractions separately until just before derivatization?

A2) We used 2-steps extraction method in order to efficient isolate compounds characterized by distinct degree of polarity. Aqueous fraction contained polar compounds and organic phase contained nonpolar compounds. We combined these two fractions (they didn’t form two phases and didn’t have effect on evaporation) prior derivatization in order to equalize the sample composition - all polar compounds converted into the non-polar derivatives. For details please see our work, where we described these method in details (Wojakowska et al., 2015; DOI: 10.1371/journal.pone.0136902)

Q3) Did the authors measure the degree of efficiency for the derivatization? Was it 100%? Were some compounds not derivatized?

A3) We have long experience in GC/MS analysis, we optimized used analytical method. The efficiency of derivatization was close to 100%. There were not derivatized compounds namely non-polar, volatile compounds without polar groups (OH, NH2, COOH).

Q 4) The authors need to explain better the quantification analysis in sections 2.5 and 2.6.  Did they use relative or absolute quantification?

A4) The corrections have been made in the text - lines 145-146.

Q5) Can the authors offer an explanation as to why they were not successful in identifying the 331  analytes other than stating that “the analytes could not be identified with implemented  approach”? (Lines 155-156). How many of these 331 analytes were found to be differentially  significant between KC and HC?  If there were significant analytes among these 331 they should  have included them in the subsequent analyses.  

A5) This is typical problem/limit in GC/MS analysis, that many of detected compounds were not identified (analytes/unknown compounds). This results from the insufficient number of compounds in databases and imperfect deconvolution algorithms. Some of analytes were found to be differentiated, but without identification of these compounds we couldn’t infer their biological role in the pathogenesis process of KC.

Q6) The PCA plot in Figure 1A does not show separation of KC from HC.  Please look at the  components of each component (PCA-1 and PCA-2). The plot shows biological variation among  samples, which is predictable, however it does not indicate clear separation.  Did the authors try and average the KC and HC samples to get a better indication of the separation?

Figure 1A does not show clear separation as Reviewer 1 suggested, but some separation which we stated in lines 173-175: "The Principal Component Analysis showed some clustering of samples from both groups, which resulted in the separation of KC and HC samples".

7) In Table 1, hexadecanol does not appear to be upregulated in KC, but rather downregulated. 

Is this just a typo?

A7) The corrections were made in the text - lines 186-187.

8) The authors found 16 differentially different compounds however they only chose to use 13 for further analyses?  What is the explanation for that?

A8) The corrections have been made in the text - line 191.

For MSEA analysis we used 13 from 16 differentiating compounds, because only 13 were indexed in Human Metabolome Database (HMDB) what is required for MetaboAnalyst calculations.

9) Lines 205-206: What are the limitations of untargeted profiling implemented in this study which prevent identification of a large number of detected compounds?

A9) As we mentioned in comment A5 there are some limitations of untargeted GC/MS-based approach. This results from the insufficient number of compounds in databases and imperfect deconvolution algorithms.

10) Lines 209-211: What are the alternative metabolomics approaches which can be used to further study differences in the molecular composition of KC and HC?

A10) As we mentioned in Introduction and Discussion section, LC/MS and NMR are alternative approaches in metabolomics study, which could be used as complementary techniques. 

11) Some other minor notes: A) Lines 28-29 – please rewrite this sentence because there is no meaning as it is written. B) Line 45 – please replace “keratoconus” with “KC”. C) Line 101 – please replace “rpm” with the corresponding “x g”. D) Line 110 – please add “an” in front of “Agilent”. E) Line 117 – is this 600 °C/min or 60 °C/min? F) In Tables 1 and S1 what does “mean level” refer to?  Intensities? Please substitute appropriately.

A11) A-D) The corrections have been made in the text; E) 600 is correct; F) The corrections have been made in the tables and text - lines:152-153

Reviewer 2 Report

The research presented in "Metabolomic Signature Discriminates Normal Human Cornea from Keratoconus – a Pilot GC/MS Pilot Study" provides further evidence that keratoconus (KC) is the result of inflammatory pathways in a sub-set of patients.  It builds on previous work as the authors outline well in the Introduction and Discussion sections; however, KC is still known to be a complex condition with many factors that lead to subtly different phenotypes.  Since the results presented here were based on a relatively small study cohort, a pilot study as indicated in the title, the authors should emphasize that not all forms of KC can be definitively linked to inflammatory metabolites.  The authors could mention something to this effect in the Introduction.  Please see Galvis, V., Sherwin, T., Tello, A. et al. Keratoconus: an inflammatory disorder?. Eye 29, 843–859 (2015) for further reference material. 

Overall, this is important work and should add to our understanding of the molecular etiology of KC.    

Author Response

Reviewer 2

The research presented in "Metabolomic Signature Discriminates Normal Human Cornea from Keratoconus – a Pilot GC/MS Pilot Study" provides further evidence that keratoconus (KC) is the result of inflammatory pathways in a sub-set of patients.  It builds on previous work as the authors outline well in the Introduction and Discussion sections; however, KC is still known to be a complex condition with many factors that lead to subtly different phenotypes.  Since the results presented here were based on a relatively small study cohort, a pilot study as indicated in the title, the authors should emphasize that not all forms of KC can be definitively linked to inflammatory metabolites.  The authors could mention something to this effect in the Introduction.  Please see Galvis, V., Sherwin, T., Tello, A. et al. Keratoconus: an inflammatory disorder?. Eye 29, 843–859 (2015) for further reference material. 

Overall, this is important work and should add to our understanding of the molecular etiology of KC. 

A: The corrections have been made in the text - lines 62-71.